Characterizing the suckling behavior by video and 3D-accelerometry in humpback whale calves on a breeding ground

Ratsimbazafindranahaka Maevatiana N. maevatiana.ratsimbazafindranahaka@universite-paris-saclay.fr 1 2 3
Huetz Chloé 2
Andrianarimisa Aristide 3
Reidenberg Joy S. 4
Saloma Anjara 1
Adam Olivier 2 5
Charrier Isabelle 2
1 Association Cétamada , Barachois Sainte Marie , Madagascar
2 Institut des Neurosciences Paris-Saclay, Université Paris-Saclay, CNRS , Saclay , France
3 Département de Zoologie et Biodiversité Animale, Université d’Antananarivo , Antananarivo , Madagascar
4 Center for Anatomy and Functional Morphology, Icahn School of Medicine at Mount Sinai , New York , United States of America
5 Institut Jean Le Rond d’Alembert, Sorbonne Université , Paris , France
McElligott Alan
Electronic publication date: 2022 Feb 17
Publication date: 2022
Volume: 10
Electronic Location ID: e12945
Received 2021 Oct 8; Accepted 2022 Jan 25
Copyright: ©2022 Ratsimbazafindranahaka et al.
Copyright year: 2022
Copyright holder: Ratsimbazafindranahaka et al.
License: This is an open access article distributed under the terms of the Creative Commons Attribution License, which permits unrestricted use, distribution, reproduction and adaptation in any medium and for any purpose provided that it is properly attributed. For attribution, the original author(s), title, publication source (PeerJ) and either DOI or URL of the article must be cited.
License URL: https://creativecommons.org/licenses/by/4.0/

Keywords: Automatic identification, Breeding area, Mother-calf interaction, Multi-sensor tag, Nursing, Suckling

Funding: The CNRS and the Cétamada association The IDEX Paris-Saclay ANR-11-IDEX-0003-02 This work was supported by the CNRS and the Cétamada association. Maevatiana N. Ratsimbazafindranahaka received a cotutelle PhD scholarship from the ADI2020 project funded by the IDEX Paris-Saclay, ANR-11-IDEX-0003-02. The funders had no role in study design, data collection and analysis, decision to publish, or preparation of the manuscript.

==============================
Getting maternal milk through nursing is vital for all newborn mammals. Despite its importance, nursing has been poorly documented in humpback whales (Megaptera novaeangliae). Nursing is difficult to observe underwater without disturbing the whales and is usually impossible to observe from a ship. We attempted to observe nursing from the calf’s perspective by placing CATS cam tags on three humpback whale calves in the Sainte Marie channel, Madagascar, Indian Ocean, during the breeding seasons. CATS cam tags are animal-borne multi-sensor tags equipped with a video camera, a hydrophone, and several auxiliary sensors (including a 3-axis accelerometer, a 3-axis magnetometer, and a depth sensor). The use of multi-sensor tags minimized potential disturbance from human presence. A total of 10.52 h of video recordings were collected with the corresponding auxiliary data. Video recordings were manually analyzed and correlated with the auxiliary data, allowing us to extract different kinematic features including the depth rate, speed, Fluke Stroke Rate (FSR), Overall Body Dynamic Acceleration (ODBA), pitch, roll, and roll rate. We found that suckling events lasted 18.8 ± 8.8 s on average (N = 34) and were performed mostly during dives. Suckling events represented 1.7% of the total observation time. During suckling, the calves were visually estimated to be at a 30–45° pitch angle relative to the midline of their mother’s body and were always observed rolling either to the right or to the left. In our auxiliary dataset, we confirmed that suckling behavior was primarily characterized by a high average absolute roll and additionally we also found that it was likely characterized by a high average FSR and a low average speed. Kinematic features were used for supervised machine learning in order to subsequently detect suckling behavior automatically. Our study is a proof of method on which future investigations can build upon. It opens new opportunities for further investigation of suckling behavior in humpback whales and the baleen whale species.

Introduction

Female mammals feed their offspring with maternal milk to ensure their progeny’s development and survival during the early dependent life stages of the young (Pond, 1977; Smith, 1977; Daly, 1979; Gittleman & Thompson, 1988; Balshine, 2012). The maternal behavior associated to the transfer of milk to the offspring is referred to as ‘nursing.’ From the offspring’s perspective, this behavior can be referred to as ‘suckling,’ i.e., to obtain milk from the mammary gland (breast) through the mammary papilla (nipple or teat) (Hall, Hudson & Brake, 1988). Suckling can involve an active role of mouth movements from the young to stimulate nipple erection, compress the breast’s lactiferous sinuses, and generate negative pressure (suction) along with a pulling motion on the teat in order to withdraw the milk. Alternatively, suckling can involve a passive role in which milk is squirted into the open mouth with no stimulating movements of the offspring’s mouth.

Nursing behavior is an integral part of mammalian reproductive behavior (Gittleman & Thompson, 1988). The duration of lactation (the period of time during which milk can be produced; Hall, Hudson & Brake, 1988) and the frequency of nursing vary greatly among mammals and are related to the developmental characteristics of the young and their environmental constraints (Oftedal, Boness & Tedman, 1987; Oftedal, 1993).

Whale breasts are positioned ventro-laterally on either side of the midline genital slit (opening of the vestibule vagina). The location of each breast is marked by a short mammary slit in the overlying blubber through which the nipple can be extruded for nursing. The mammary glandular tissue is located deep to the overlying blubber layer, thus providing a hydrodynamically streamlined outline even when the whale’s breasts are engorged with milk during lactation. Lactation in baleen whales typically occurs initially in the winter, when calves are born (Lockyer, 1984; Oftedal, 1993; Bannister, 2018). As most baleen whales migrate, lactation that begins on the calving area continues during the migration to the feeding area (Lockyer, 1984; Oftedal, 1993; Bannister, 2018).

Humpback whales (Megaptera novaeangliae), the subject of this study, are migratory cetaceans. They move between high-latitude (closer to the poles) feeding grounds and low-latitude (closer to the equator) breeding/calving grounds (Clapham, 2018). Similar to other baleen whales, their nursing strategy is constrained by a prolonged fasting of the mother during lactating period, related to the migratory pattern (Oftedal, 1993). A young humpback whale (calf) feeds exclusively on maternal milk during its first six months of life (Clapham, 2018). The energy intake from the milk is crucial in ensuring rapid growth (Lockyer, 1984) and therefore allowing the calf to migrate to the feeding area in cold waters alongside its mother (Lockyer, 1984; Corkeron & Connor, 1999; Braithwaite, Meeuwig & Hipsey, 2015). Mother-calf pairs make slower progress compared to groups not accompanied by calf in part due to presumed slower swimming or frequent pauses that allow the calf to suckle (Noad & Cato, 2007). After its first six months of life, the calf starts to feed on prey but continues to get nutrition from milk until complete weaning at 10 to 12 months old (Chittleborough, 1958; Clapham, 2018). The weaning coincides with the mother returning to the calving area. Although early weaning on the feeding grounds may also occur (Baraff & Weinrich, 1993; Steiger & Calambokidis, 2000).

Despite its importance, nursing (and suckling) behavior has been poorly documented in humpback whales because it is difficult to observe with accuracy and certainty. Indeed, nursing can occur at varying depths from the sub-surface (<5 m) to deep dives (up to 64 m) as suggested by camera-observed nursing events in feeding areas (Tackaberry et al., 2020). The earlier descriptions of nursing behavior in humpback whales were mostly based on limited surface and sub-surface observations of stationary or slow moving whales and relied mainly on the positioning of the calf (Glockner & Venus, 1983; Glockner-Ferrari & Ferrari, 1985; Clapham & Mayo, 1987; Morete et al., 2003; Videsen et al., 2017). Clapham & Mayo (1987) for example described nursing as an event between two calf’s successive breaths on either sides of the mother’s caudal peduncle when the mother is stationary at the surface and during which the calf is beneath the mother. Videsen et al. (2017) defined nursing events by associating them with the occurrence of ‘peduncle dives,’ defined as repeated dives during which the calf remained submerged beneath the mother’s caudal peduncle.

Nevertheless, from these different studies, various nursing modalities have been advanced. While nursing, the mother is either stationary or swimming very slowly at a depth around 10–15 m (Glockner-Ferrari & Ferrari, 1985), or occasionally at the surface with her tail in the air (Morete et al., 2003). The calf is typically positioned vertically (head up, tail down) beneath the mother (Glockner & Venus, 1983). Videsen et al. (2017) suggested that to initiate suckling, the calf uses tactile stimuli on the mother rather than vocalizations. Tactile stimulation likely results in an autonomic reflex causing extrusion of the nipple. The calf can then grasp the nipple with its tongue. From an underwater video footage of suckling humpback whale calf recorded in the tropical waters of Tonga, South Pacific Ocean, by Scott Portelli, 2021, pers. comm., it has been observed that the whale calf has an unusual tongue that is lined with lingual papillae (fringes) that may help the tongue grasp the nipple, and may even seal the tongue into a tube that facilitates transferring milk into the mouth.

Underwater video is the best method for studying the nursing and suckling behaviors of whales. Various solutions are possible. One option is to use a video camera from a surface platform (boat or kayak) but it has two major drawbacks: (1) it is necessary to get very close to the whales and potentially interfere with their behavior (harassment); (2) visual contact is lost when the whales dive deeply, even when the water is very clear. Another option is to use an underwater drone (Remotely Operated underwater Vehicle—ROV or Autonomous Underwater Vehicle—AUV, Butcher et al., 2021). ROVs and AUVs have potential but presently vehicles are not yet agile enough to track whales. They are too slow and difficult to manoeuver, with ROVs also being limited by cable length. Alternatively, underwater video can be obtained during close approaches by breath-holding or scuba divers. However, this is only possible when the whales are stationary or swimming very slowly as it would be impossible for a diver to keep pace with traveling whales. Additionally, the presence of divers in close proximity may disturb the whales and habituation is not a desirable tactic. Scott Portelli, 2021, pers. comm., Zoidis & Lomac-MacNair (2017) successfully video documented humpback whale suckling behavior thanks to breath-holding divers. In Zoidis & Lomac-MacNair (2017), the recording of milk clouds in the water column allowed the confirmation of the occurrence of suckling. However, only few events could be recorded (N = 5 from four mother-calf pairs).

The development of animal-borne multi-sensor tags such as Customized Animal Tracking Solutions (CATS) cam tags provided a way for remote and accurate recording of whale behavior (Cade et al., 2016). Bio-loggers can be equipped with a suite of sensors including cameras, accelerometers, gyroscopes, magnetometers, pressure (depth) sensor, temperature sensor, light sensor, and hydrophone. CATS cam tags are particularly well suited to study suckling behavior since in addition to kinematic sensors they also allow the recording of video directly from the calf’s perspective and thus allow confident confirmation of suckling events. Such devices were used by Tackaberry et al. (2020) to accurately describe the suckling behavior of humpback whale calves on feeding areas. One disadvantage of using an animal-borne tag is the potential stress generated during the tag attachment phase. The deployment must be done by experienced operators following a strict protocol (slow speed approach, brief deployment time, etc.) to minimize the disturbance (Stimpert et al., 2012; Saloma, 2018).

In this study, we characterized the suckling behavior of <3 months old humpback whale calves on their calving ground using CATS cam tags. We aimed to describe in detail: (1) how and at what depth the suckling is performed, its duration, and its frequency; (2) the behavioral signatures of suckling events using accelerometry and depth data; and (3) whether the different kinematic features extracted from only two sensors (i.e., 3-axis accelerometer and depth sensor) among all the available sensors in the CATS cam tags were sufficient for a supervised machine learning algorithm to automatically detect suckling behavior. Supervised machine learning is a type of machine learning in which an algorithm learns from labelled datasets to classify unforeseen data. Such classification technique would open the opportunity to detect suckling events from data collected by tags without video camera such as Acousonde or Dtags or when visual observations are not good enough (e.g., during night time or in water with poor visibility).

Materials & Methods

Study area

The study occurred in the Indian Ocean along coast of Madagascar, specifically in the Sainte Marie channel. The channel is located between the Sainte Marie Island (between latitudes 17°19′and 16°42′South, and longitudes 49°48′and 50°01′East) and the east coast of Madagascar’s mainland. The channel is approximately 60 km long and 7–30 km wide. It is relatively shallow: its average depth is about 35 m and the maximal depth is 60 m (Trudelle et al., 2016). The tag deployments were conducted as part of an ongoing study on humpback whale mother-calf interactions during the calving season and occurred between August and September of 2018 and 2019.

Tag specifications

We used CATS cam tags to investigate humpback whale suckling behavior. CATS cam tags are small and lightweight (∼500 g) non-invasive animal-borne multi-sensor tags attached via suction cups. They contain six auxiliary sensors (3-axis accelerometer, 3-axis gyroscope, 3-axis magnetometer, pressure/depth sensor, temperature sensor, and light sensor), a hydrophone, and an HD video camera (100° Field of view). A VHF transmitter (ATS F1835B) attached on the tag allows tracking for tag retrieval. The sampling frequency was set at 10 Hz for the magnetometer, gyroscope, depth, and temperature sensors. Accelerometer sampling rate was set at 400 Hz in 2018 and at 800 Hz in 2019. Accelerometer data from 2019 were however always downsampled to 400 Hz in subsequent analyses for a matter of consistency. The camera recorded videos with a 1,280 × 720-pixel resolution at 30 frames per second in 2018 and with a 1,920 × 1,072-pixel resolution at 30 frames per second in 2019. The hydrophone recorded sound at a 48 kHz sampling rate in 2018 and at a 24 kHz sampling rate in 2019 (16-bit resolution). Two individual CATS cam tags were used in our study (designated as Cats1 and Cats2 hereafter).

Tagging procedures

CATS cam tags were deployed on calves accompanied by their mother from a 6.40 m rigid motor boat using a 5-m carbon-fiber pole. Deployment were performed by researchers experienced in successfully approaching mother-calf pairs with minimal disturbance. The tags were placed on the back, near the dorsal fin of the animal. Calves were tagged using one of the two approaches described in Stimpert et al. (2012) and in Saloma (2018) in order to minimize disturbance to the mother-calf pair. Tagging efforts were terminated if the pair displayed avoidance behavior or if the calf was not successfully tagged within 30 min. Immediate behavioral response of the animals to tagging was recorded as in Stimpert et al. (2012). All mother-calf pairs were photo-identified to avoid double-sampling within the calving season. We attributed a relative age to each tagged calf depending on the angle of furl of the dorsal fin (neonate versus non-neonate, Cartwright & Sullivan, 2009; Faria et al., 2013; Saloma, 2018).

Tagged animals were not followed after tag deployment to avoid any further disturbance of their behavior. After tag deployment or an aborted attempt, the boat slowly moved away in the opposite direction of the mother-calf pair. The tag was retrieved after few hours or the following day, when it detached itself from the animal (usually as a consequence of rubbing against the mother, surface active behavior, etc.). The VHF tag emitted a continuous signal, facilitating retrieval. All methods and approaches were carried out in accordance with relevant guidelines and regulations in force in Madagascar and were approved by the Ministry of Fisheries Resources, Madagascar, under the national research and collect permits #28/18-MRHP/SG/DGRHP and #36/19-MAEP/SG/DGPA. This present study complies with the European Union Directive on the Protection of Animals Used for Scientific Purposes (EU Directive 2010/63/EU) and also with current Malagasy laws.

Sensor data processing

Data from all sensors were downloaded as CSV files and imported into MATLAB (Mathworks) using dedicated scripts (CATS Matlab toolkit, https://github.com/wgough/CATS-Methods-Materials, Cade et al., 2021). Raw accelerometer data were downsampled to obtain a common sampling rate of 10 Hz across all sensors and depth data (in meters) was smoothed with a 0.5 s running median filter. In all computation described hereafter the output sampling frequency was of 10 Hz. Sensors reading were rotated to match the calf’s orientation frame using established methods and raw animal pitch and roll (in degree) were then calculated (Johnson & Tyack, 2003; Cade et al., 2016; Tackaberry et al., 2020; Cade et al., 2021). To obtain pitch and roll data representing only the body posture and without fluke stroke signal, we low-pass filtered the raw pitch and roll data (0.2 Hz low-pass filter) as in Simon, Johnson & Madsen (2012). On the other hand, we used band-pass filtered raw pitch (0.2–1 Hz band-pass filter) to obtain the fluke stroke pattern, similar to Simon, Johnson & Madsen (2012). Stroking was identified when the band-pass filtered pitch passed from below −3° to above +3° or vice versa within 6.5 s. The used thresholds were determined by visual inspection of the accelerometry plots (Iwata et al., 2021). From the identified fluke strokes, the Fluke Stroke Rate (FSR, in Hertz) was calculated on the basis of half-strokes (López et al., 2015). Animal forward speed (speed hereafter, in meter/second) was determined using the tag jiggle recorded in the original high frequency accelerometer data (Cade et al., 2017). Speed was smoothed with a 0.5 s running mean filter. In addition, we calculated the Overall Dynamic Body Acceleration (ODBA, in G and converted to meter/second2) as in Wilson et al. (2006), the roll rate (in degree/second), and the depth rate (in meter/second) using custom scripts and the Animal Tag toolbox (http://www.animaltags.org).

Depth data helped determine various diving or surface activity phases. We defined diving as any submergence to a depth of >10 m (Stimpert et al., 2012; Saloma, 2018). Dives were further divided into three phases: descent, bottom, and ascent phase. As individual dives could include stops at various depths, the bottom phase was defined as the segment at >85% of the maximal dive depth for a dive (Stimpert et al., 2012). The descent phase was defined as the segment starting at the surface that immediately preceded the bottom phase. Inversely, the ascent phase was defined as the segment that follows directly the bottom phase and ends at the surface.

Video data analysis and suckling data extraction

We identified and labelled suckling events from the video files using the Behavioral Observation Research Interactive Software (BORIS; Friard & Gamba, 2016). The corresponding depth data was displayed concurrent with the video. We defined a suckling event as a period during which the tip of the calf’s snout continuously touched (>2 s) the mammary slit of the mother and a milk cloud (see video clip in https://www.youtube.com/watch?v=UcyCgiCieFk), even in low density, was observed in the water during the event or upon release of contact (Tackaberry et al., 2020). For each suckling event, we calculated its duration and extracted its corresponding activity phase (descent, bottom, ascent, or surface), together with the average depth, depth rate, speed, FSR, ODBA pitch, roll, and roll rate. In addition to suckling events identification, we estimated the proportion of time each calf was in close proximity under its mother (mother visible above the calf) without suckling. Furthermore, we also counted the number of suckling dives during which one or several confirmed suckling events were recorded and the number of non-suckling dives during which the calf is observed staying at least 5 s under the mother, in close proximity and without any suckling event.

Comparison of suckling with non-suckling segments

A reference data set for comparison is needed to contrast and identify the behavioral signatures unique to suckling events. To generate such comparison data, we divided our dataset from each deployment into non-overlapping segments of 20 s (duration comparable to suckling events). Then we selected 10 random segments that did not match to any suckling event for each activity phase in which suckling occurred, similar to Tackaberry et al. (2020). These selected segments, referred as ‘non-suckling segments,’ were analyzed for the same characteristics as suckling events (see above). To assess if there was a significant difference between suckling events and non-suckling segments, we used a linear mixed-effects models (LMMs) estimated in R using REML (Restricted Maximum Likelihood) that included the suckling status (suckling versus non-suckling) and the activity phase as fixed effects, and individuals as random effect (Tackaberry et al., 2020). In the models, we considered the following response variables: average depth rate, average speed, average FSR, average ODBA, average pitch, absolute average roll, and average roll rate. Absolute value has been used for roll to emphasize any deviation from zero. The model’s reference levels (intercept) corresponded to bottom non-suckling (non-suckling segment occurring at the bottom phase of dive). The LMMs were followed by Tukey’s post-hoc multiple comparison test using the R package emmeans (Lenth et al., 2018). Statistical significance level was set to α = 0.05.

Testing automatic identification of suckling behavior using supervised machine learning

Data preparation

For the supervised machine learning process, as the goal was to identify automatically suckling in non-labelled data, the entire dataset was included in the analysis (i.e., data with video from tag on to tag off). For each deployment, we split the dataset into 2 s non-overlapping blocks (windows) and then determined which behavioral period each block fell under (suckling or non-suckling period). Two classes were thus considered: (1) ‘suckling’ and (2) ‘non-suckling.’ Each block falling under both suckling and non-suckling period (i.e., transition to or from a suckling period) was labelled as ‘suckling’ only if 3/5 of its duration fell under suckling period. Otherwise, it was labelled as ‘non-suckling.’ For each block, we computed 43 features: mean, minimum, maximum, variance, skewness and kurtosis for depth, depth rate, speed, ODBA, pitch, roll, and roll rate, and finally mean for FSR. Indeed, as suggested in Ladds et al. (2016), a great number of summary statistics may help in detecting subtle differences between classes.

Similar to Jeantet et al. (2020), we segmented the data prior to machine learning implementation. We excluded blocks too close to the surface (<1.5 m depth) or of high speed activities (>2 m s−1). This exclusion helped in partially removing noise from the data and also in reducing class imbalance. As suggested by observations on a feeding ground, suckling events occur mostly at depth (Tackaberry et al., 2020). Also, given the size of the mothers (>11 m in length and >2 m diameter at the umbilical section level in adult whales, as determined by estimates in the field), the generally observed nursing configuration (calf generally below the mother), and speed during nursing (Zoidis & Lomac-MacNair, 2017; Tackaberry et al., 2020), we were confident that no suckling events would be recorded at <1.5 m depth or at >2 m s−1 speed. We checked the validity of these threshold assumptions in our data to ensure that indeed no suckling events were removed partially or entirely. Validating these parameters ensured that this segmentation could be automatically performed on unknown data, with a very low probability to remove suckling events.

Identification of the adequate classifier

In order to define an appropriate classification model (machine learning model) for associating the suckling status of the calf with the corresponding patterns of different kinematic features, we trained four types of supervised machine learning algorithms —(i) K-Nearest Neighbors (KNN), (ii) Decision tree, (iii) Ensemble classifiers, and (iv) Support Vector Machine (SVM)—using the MATLAB toolbox Statistics and Machine Learning. We repeatedly performed a 60:40 holdout splits on our data (i.e., 60% of the data as training set and 40% of the data as testing set) while maintaining class ratios (30 runs): the training set was used for training and the classifier’s efficiency was evaluated on the remaining unseen data. The holdout split is a commonly used approach for evaluating general model performance in many machine-learning applications (e.g., Nathan et al., 2012; Carroll et al., 2014; Ladds et al., 2016). Preliminary model selection and hyperparameter tuning for each classifier type were performed using a Bayesian Optimization approach with 5-fold cross-validation. We followed the workflow described in the associated documentation (https://fr.mathworks.com/help/stats/bayesian-optimization-workflow). With the Statistics and Machine Learning toolbox’s Bayesian Optimization algorithm, we optimized the different types of supervised machine learning algorithms across a selection of classification models and hyperparameter values in order to pre-select the best classification models that were suitable for our dataset. An optimization run pre-selected one model as the best one by seeking to minimize classification error. For Ensemble classifiers, the selection of classification models included Bootstrap Aggregation (Bagging), Random Forest (Bag), Random Subspace (Subspace), Adaptive Boosting for Binary Classification (AdaBoostM1), Adaptive Boosting for Multiclass Classification (AdaBoostM2), Gentle Adaptive Boosting (GentleBoost), Adaptive Logistic Regression (LogitBoost), Linear Programming Boosting (LPBoost), Least-Squares Boosting (LSBoost), Robust Boosting (RobustBoost), Random Undersampling Boosting (RUSBoost), and Totally Corrective Boosting (TotalBoost). For KNN classifiers, it included Cityblock, Chebyshev, Correlation, Cosine, Euclidean, Hamming, Jaccard, Mahalanobis, Minkowski, Seuclidean, and Spearman metrics-based KNN. For Decision trees, it included classifiers that are based on Gini’s diversity index (Gdi), twoing rule, and deviance splitting criterion. For SVM, it included SVM with Gaussian, linear, and polynomial kernels. These different models are detailed in the aforementioned documentation. The number of runs we used for the data splitting (30 runs) and for the Bayesian optimization (100 runs per split per classifier type) were chosen to optimize results stability and processing speed.

For each trial and each class, we calculated five evaluation metrics for making a final decision on the classification model to be retained: Sensitivity (True Positive Rate, Hit Rate or Recall), Precision (Positive Predictive Value), False Positive Rate (FPR), F-score, and Global accuracy. These metrics were calculated as follows and used to select the best and most adapted classification model (definitions as in Jeantet et al., 2020):

The Sensitivity measures the ability to detect one behavior among other behaviors: (1) Sensitivity=TPTP+FN.

The Precision measures the ability to correctly identify a behavior: (2) Precision=TPTP+FP.

The FPR measures the rate of wrongly considering other behaviors as the behavior of interest. It is related to the Specificity, which is the ability to avoid wrongly considering other behaviors as the behavior of interest:

(3) FPR=1−SpecificitywithSpecificity=TNTN+FP

The F-score measures the accuracy in classifying a behavior. It is the harmonic mean of precision and sensitivity: (4) F−score=2TP2TP+FP+FN

The Global accuracy measures the ability to correctly identify all behaviors as a whole: (5) Globalaccuracy=TP+TNTP+TN+FP+FN.

These formulas use the following abbreviations: TP–True Positive, TN–True Negative, FN–False Negative), and FP–False Positive.

To assess the potential influence of training set size reduction on the performance of the elected model, we tested whether changing the training set size changed the performance. We ran a series of trainings with different holdout splitting of the data (training-testing): 60:40 as in the original model selection process, 50:50, 40:60, 30:70, 20:80, and 10:90. The process was repeated 30 times for each splitting. A general workflow of the whole machine learning process we followed is presented in Fig. S1.

Generalization across individuals

To assess whether the model can generalize across different individual whale calves, we also tested a leave-one-out split, similar to Ladds et al. (2016). We trained the model with the highest performance on two calves’ data. We then tested its performance on the unseen remaining calf data, establishing if the tag data obtained from individuals could be used to identify the suckling behavior of a different individual. We repeated this analysis three times, leaving out a different calf’s data each time following the same machine learning workflow outlined in Fig. S1.

Aside from possible influence of training set size (e.g., the recorded data from the individuals used for training is relatively small), lack of generalization ability across individuals may result from a dataset with high inter-individual variability of suckling behavior characteristics. To address this potential issue, we tested whether excluding features that contribute most to inter-individual difference (if any) in the suckling blocks would improve the performance of the supervised machine learning. To identify the features that contribute most to inter-individual difference, we trained a Random Forest algorithm to classify individuals based only on the suckling blocks using the R software package randomForest (Liaw & Wiener, 2002). Features that contributed the most to the differentiation by individual were identified as those that had a high Gini index in the Random Forest. We re-ran the leave-one-out design, as described above, but excluding the features identified by using the Gini index. We confirmed the reduction of the inter-individual difference of the suckling blocks by examining the error rate (Out-of-the-bag error rate) of the Random Forest algorithm trained to classify individuals. A reduction of inter-individual difference would result in an increase of the error rate. A low error rate indicated that the inter-individual difference is evident enough and thus the Random Forest algorithm is able to make very good classifications.

Results

Tag deployments

Four calves were tagged with CATS cam tag in the Sainte Marie channel: one during the calving season of 2018 (identified as Calf1) and three during the calving season of 2019 (identified as Calf2, Calf3, and Calf4). Calf1 and Calf2 were tagged with Cats1 while Calf3 and Calf4 were tagged with Cats2. Their immediate reaction to tagging was a slow evasive swimming, which corresponded to a mild reaction (Stimpert et al., 2012). All of them had unfurled dorsal fins, indicating that they were not neonate calves (Cartwright & Sullivan, 2009). The deployment on Calf4 was not suitable for our study as the tag (thus the camera) was pointing toward the side of the calf (i.e., away from the mouth) and was thus excluded from our analyses. For Calf3, the data towards the end of the deployment was not usable and thus not analyzed due to lack of visibility on the corresponding video recording as the night approached. Deployment information including date and time is indicated in Table 1. The three deployments provided a total of 10.52 h of usable video and auxiliary data.

Table 1 Characteristics of all suckling events for each calf.

Date-times are presented in the format DD/MM/YYYY hhmm–hhmm and corresponds to deployment date, data start time, and data end time. Data start time also corresponds to tag attachment time. Data end time corresponds to tag detachment time, except for Calf3 since the data towards the end of the deployment on Calf3 was not usable due to lack of visibility on the video recording as the evening approached (the tag detached at 1845). The letters R/L indicate the rolling sides of the calf as observed on the video recordings: Right/Left. Budget represents the proportion of time the animal was observed suckling. The remaining values are presented following the format mean ± SD (min; max). Note the use of absolute value for roll to emphasize the roll deviation from zero (regardless of the side). No suckling events were observed during ascent phase of dives.

						Averages	
ID	Date-time	Occurrence	R/L	Duration
(s)	Budget
(%)	Depth
(m)	Depth rate
(m s −1 )	Speed
(m s −1 )	FSR
(Hz)	Pitch
(°)	—Roll—
(°)	Roll rate
(°s−1)	ODBA
(m s −2 )	
Calf1	14/09/2018
1134–1418	During dive
(descent)
N = 2	1/1	23.3 ± 1.5
(22.2; 24.3)	0.5	24.6 ± 3.3
(22.3; 26.9)	0.2 ± 0.1
(0.1; 0.2)	1.5 ± 0.1
(1.4; 1.5)	0.3
(0.3; 0.3)	−13 ± 12
(−22; −5)	39 ± 13
(30; 48)	3.6 ± 1.9
(2.2; 4.9)	0.4
(0.3; 0.4)	
During dive
(bottom)
N = 3	2/1	22.2 ± 12.8
(8.8; 34.2)	0.7	30.6 ± 1.2
(29.3; 31.8)	0.1 ± 0.3
(−0.2; 0.4)	1.4 ± 0.1
(1.4; 1.5)	0.2
(0.1; 0.2)	−5 ± 14
(−15; 12)	37 ± 15
(20; 49)	4.1 ± 0.8
(3.5; 5.1)	0.4 ± 0.1
(0.3; 0.5)	
Calf2	06/08/2019
1057–1234	During dive
(descent)
N = 1	0/1	44.9	0.8	22.2	0.1	1.6	0.1	25	63	3.5	0.4	
During dive
(bottom)
N = 3	3/0	18.4 ± 12.3 (6.2; 30.9)	0.9	26.1 ± 6.7
(19.3; 32.6)	0 ± 0.2
(−0.1; 0.3)	1.7 ± 0.1
(1.6; 1.8)	0.2
(0.2; 0.2)	11 ± 7
(3; 16)	47 ± 3
(44; 48)	3.7 ± 0.9
(3.1; 4.8)	0.5 ± 0.1
(0.5; 0.6)	
Calf3	09/08/2019
1003–1802	During dive
(descent)
N = 6	2/4	10.9 ± 4.2
(4.8; 15.6)	0.3	12.4 ± 3.4
(8.9; 17.8)	0.2 ± 0.1
(0.1; 0.3)	1.3 ± 0.1
(1.2; 1.5)	0.3 ± 0.1
(0; 0.4)	3 ± 10
(−10; 19)	47 ± 12
(27; 63)	5.8 ± 2.7
(3.1; 9.9)	0.3 ± 0.1
(0.2; 0.4)	
During dive
(bottom)
N = 14	8/6	18.9 ± 4.6
(11.2; 25.5)	1.1	16.8 ± 3.2
(11.1; 21.1)	0 ± 0.1
(−0.1; 0.3)	1.3 ± 0.2
(1.1; 1.7)	0.3 ± 0.1
(0; 0.4)	10 ± 7
(0; 25)	51 ± 9
(25; 61)	4.6 ± 2.1
(0.5; 8.6)	0.3 ± 0.1
(0.3; 0.4)	
At surface
N = 5	3/2	19.6 ± 10
(7.9; 35)	0.4	6.2 ± 2.7
(2; 8.4)	0 ± 0.1
(−0.1; 0.1)	1.2
(1.2; 1.2)	0.2 ± 0.1
(0.2; 0.3)	5 ± 10
(-6; 19)	37 ± 24
(6; 59)	6.4 ± 1.9
(4.5; 9)	0.3 ± 0.1
(0.2; 0.4)	

General description of suckling

In all our video recordings, a milk cloud could be observed during or after contact each time the calf’s snout continuously touched (>2 s) the mammary slit. All these events appeared to be intentional and were all defined as suckling. We detected 34 suckling events, which lasted on average 18.8 ± 8.8 s and represented 1.7% of the total video deployment (Fig. S2). Detailed summary statistics per individual are presented in Table 1. Most suckling events were clustered in a series of 2–6 suckling events occurring less than a minute apart (Fig. S2). Eighty-eight percent of the suckling events (30 suckling events) were recorded during ten different clusters of suckling events. The global temporal distribution of the suckling events through the duration of each deployment are presented in Fig. S3. There was no evidence of increasing trends in suckling frequency through the duration of the deployments (no indication of a habituation effect, i.e., habituation to tag). Indeed, for the three deployments the first recorded suckling events occurred within the first 90 min of the deployment (within the first 30 min for Calf2) and the periods towards the end of the deployments were not necessarily associated with more frequent suckling events (Fig. S3).

The majority of suckling events occurred while diving, during the descent and the bottom phases (N = 9 for descent suckling and N = 20 for bottom suckling, Table 1). Suckling during descent and during bottom phases of dives occurred at a mean average depth of 16.2 ± 6.5 m and 20.2 ± 6.6 m respectively (Table 1). Only five suckling events occurred at the surface, at a mean average depth of 6.2 ± 2.7 m (Table 1). No suckling event occurred during ascent phase of dives.

While calves were visually estimated to be at about 30–45° pitch angle relative to the midline of their mother’s body during suckling in the videos, the pitch recorded by the tag (pitch angle relative to the horizontal) was fairly lower on average (during descent: mean average = 2 ± 14°, bottom: mean average = 8 ± 10°, at surface: mean average = 5 ± 10°, Table 1). Visual estimation of the calf’s pitch assumed that the mother had a relatively straight body during nursing, as we observed no apparent extreme body arching of the mother. In all of the observed events, we could visually tell that the calf was always rolling to one side (to the right or to the left). These rolling periods during suckling were recorded by the tag as a sustained deviation of the roll from zero during the event (Fig. S2). In all but one cluster of suckling events, the calf alternated between suckling with a right roll then rotating to suckle with a left roll or vice-versa (Fig. S2). In eleven suckling events (one for Calf1, four for Calf2 and six for Calf3), we were able to visually confirm that the calf’s mouth covered only one mammary teat during a suckling event. In three of these events, the tongue was clearly visible and displayed a rhythmic movement (Fig. 1, see also example footage in https://www.youtube.com/watch?v=nyqhb9BemYI). For these events with confirmed covered teat, rolling to the right corresponded to a mouth covering the right teat (N = 3) and rolling to the left corresponded to a mouth covering the left teat (N = 8, Fig. S2).

Figure 1 Screen capture of a video footage of humpback whale calf suckling on the right teat of its mother.

Photo credit: Isabelle Charrier.

Relation between calf’s positioning and suckling during dive

During our tag deployments, Calf1, Calf2, and Calf3 spent respectively 11.7%, 12%, and 8.9% of their total video time in close proximity to their mother’s ventral side without suckling. Non-suckling dives in which the calf remained beneath the mother for at least consecutive 5 s were more common than suckling dives (Table S1). In other words, using calf proximity to the mother’s ventral side alone does not indicate that suckling is occurring. Additional accelerometry cues, which are described below, are thus necessary to determine the occurrence of this behavior.

Behavioral signatures of suckling

The characteristics of suckling events and non-suckling segments are summarized in Fig. 2. Raw data are provided in Data S1. The detailed results of the LMMs are presented in Table S2 and the detailed results of the post-hoc tests are provided in Table S3. Within our model, for the average depth rate, the effect of suckling, surface phase, and the interaction effect between surface phase and suckling were all statistically non-significant and close to zero (Suckling: β = −0.01, P = 0.85, Surface: β = −0.02, P = 0.849, Suckling*Surface: β = 0.03, P = 0.863, Table S2). The effect of descent was on the other hand statistically significant, substantial, and positive (β = 0.6, P < 0.001, Table S2). The interaction effect between suckling and descent was also statistically significant and was moderate, although negative (β = −0.44, P < 0.001, Table S2). In other words, descent phase was associated with a significant increase in average depth rate, which is expected, but the increase is significantly attenuated when suckling. Bottom phase and surface phase were associated to comparable low average depth rate and were not significantly influenced by the act of suckling. There was substantial and statistically significant difference in average depth rate between suckling and non-suckling during descent (Tukey’s post-hoc test, β = −0.46, P < 0.001, Table S3). The difference in average depth rate between suckling and non-suckling at the bottom of dive and at surface were however weak and statistically non-significant (Tukey’s post-hoc test, Bottom suckling - Bottom non-suckling, β = −0.01, P = 1, Surface suckling - Surface non-suckling, β = 0.01, P = 1, Table S3).

Figure 2 Comparison of suckling events (18.8 s average duration) and non-suckling segments (20 s duration) with respect to activity phases.

Avg.: average. Mean and median are indicated by diamond marks and bold horizontal lines respectively. No ascent suckling has been observed.

With respect to average speed, the effect of descent phase and surface phase were statistically significant and positive (Descent: β = 0.22, P = 0.016, Surface: β = 0.3, P = 0.033, Table S2). The effect of suckling, the interaction effect between descent phase and suckling, and the interaction effect between surface phase and suckling were negative but were statistically non-significant (Suckling: β = −0.1, P = 0.363, Suckling*Descent: β = −0.21, P = 0.22, Suckling*Surface: β = −0.38, P = 0.082, Table S2). To be specific, descent phase and surface phase were associated with a significant increase in average speed irrespective of the suckling state. Globally, post-hoc tests showed that the average speed tended to be lower during suckling compared to non-suckling for all activity phases (Tukey’s post hoc test, negative β for all activity phase, Table S3). However, the difference was statistically non-significant (Tukey’s post-hoc test, P > 0.05, Table S3).

Suckling, descent phase and surface phase had moderate and statistically significant positive effect on average Fluke Stroke Rate (FSR) (Suckling: β = 0.16, P < 0.001, Descent: β = 0.06, P = 0.04, Surface: β = 0.1, P = 0.016, Table S2). The interaction of suckling with descent phase and with surface phase had negative but statistically non-significant effect (Suckling*Descent: β = −0.06, P = 0.258, Suckling*Surface: β = −0.12, P = 0.073, Table S2). Specifically, descent phase and surface phase were associated with an increase in average FSR and for all activity phases the act of suckling was associated with an increase in average FSR. Post-hoc tests (Table S3) showed however that there was no statistically significant difference in average FSR between suckling and non-sucking except when at bottom (Tukey’s post-hoc test, β = 0.16, P < 0.001).

Regarding the average Overall Dynamic Body Acceleration (ODBA), only descent had statistically significant effect (β = 0.16, P = 0.028, Table S2). Suckling and surface phase had statistically non-significant effect (Suckling: β = 0.05, P = 0.522, Surface: β = 0.1, P = 0.377, Table S2). The interaction of suckling with descent phase and with surface phase also had statistically non-significant effect (Suckling*Descent: β = −0.2, P = 0.132, Suckling*Surface: β = 0.1, P = 0.551, Table S2). In other words, descent phase was associated with an increase in average ODBA and for all activity phases, the act of suckling did not significantly influence the average ODBA. Overall, there was no statistically significant difference in average ODBA between suckling and non-sucking for all activity phases (Tukey’s post-hoc test, −0.15 ≤β ≤ 0.05, P > 0.05, Table S3).

For average pitch, the effect of surface phase and the interaction effect between surface phase and suckling were weak and statistically non-significant (Surface: β = −5.05, P = 0.222, Suckling*Surface: β = 1.41, P = 0.829, Table S2). The effect of suckling was weak, yet statistically significant (β = 6.36, P = 0.044, Table S2). The effect of descent phase was on the other hand statistically significant, substantial, and negative (β = −18.26, P < 0.001, Table S2). The interaction effect of suckling on descent phase was also statistically significant and moderate but positive (β = 12.77, P = 0.011, Table S2). Specifically, descent phase was associated with a significant decrease in average pitch, which is expected, but the decrease was significantly attenuated when suckling (posture close to the horizontal when suckling). Bottom phase and surface phase were associated to an average pitch close to zero, positively influenced by the act of suckling (slight upward pitch when suckling). There was substantial and statistically significant difference of average pitch between suckling and non-suckling during descent (Tukey’s post-hoc test, β = 19.13, P < 0.001, Table S3). The difference in average pitch between suckling and non-suckling at the bottom of dive and at surface were however weak and statistically non-significant (Tukey’s post-hoc test, Bottom suckling - Bottom non-suckling, β = 6.36, P = 0.345, Surface suckling - Surface non-suckling, β = 7.77, P = 0.756, Table S3).

The effect of suckling on average absolute roll was statistically significant and very strong (β = 38.46, P < 0.001, Table S2). All other effects were statistically non-significant and mostly weak (Descent: β = −0.44, P = 0.882, Surface: β = 0.64, P = 0.885, Suckling*Descent: β = −0.25, P = 0.965, Suckling*Surface: β = -11.56, P = 0.108, Table S2). In other words, all activity phases were associated with fairly low average absolute roll and the average absolute roll was positively influenced by suckling, regardless of the activity phase, i.e., the average absolute roll was high during suckling for all activity phases. Post-hoc test indicated a significant difference in average absolute roll between suckling and non-sucking for all activity phases (Tukey’s post-hoc test: 26.89 ≤ β ≤ 38.46, P ≤ 0.001, Table S3).

The effect of suckling, descent phase and surface phase, and the interaction effect between descent phase and suckling and between surface phase and suckling on the average roll rate were all low and statistically non-significant (Suckling: β = 1.17, P = 0.06, Descent: β = 0.32, P = 0.559, Surface: β = 0.28, P = 0.726, Suckling*Descent: β = 0.4, P = 0.692, Suckling*Surface: β = 1.51, P = 0.249, Table S2). The average roll rate displayed by calf during bottom phase, descent phase, and surface phase were comparable and the average roll rate was not significantly influenced by suckling. Overall, there was no statistically significant difference of roll rate between suckling and non-sucking for all activity phases (Tukey’s post-hoc test: β ≤ 2.68, P > 0.05, Table S3).

Best classifier for automatic identification of suckling

A total of 7,827 behavioral blocks of 2 s duration (4,697 for training and 3,130 for testing, 60:40 holdout splitting) were obtained from the datasets after removing blocks too close to the surface and those of high speed activities (see Segmentation in the Method section). The raw data used for this part of the study are available in Data S2. No suckling events were affected by our thresholds for removing blocks too close to the surface and those of high speed activities. The class ‘suckling’ represented 4% of the data and the class ‘non-suckling’ represented 96%. Figure 3A presents all models pre-selected by the optimization procedure for identifying automatically suckling in non-labelled data. How often each model was selected over the 30 runs of the optimization procedure for each type of classifier is presented in Fig. 3A, and details are presented in Table S4.

Figure 3 Models’ performance in automatically identifying suckling blocks.

The data included all three tag deployments on humpback whale calf. A 60:40 holdout training-testing split was used. (A) Sensitivity versus Precision plot of 12 models pre-selected using a Bayesian optimization approach. The symbols show the mean values and the SD. The models were pre-selected in the optimization procedure for each type of classifier over 30 runs. (B) Example of global results from automatic identification of suckling behaviour using the AdaBoostM1 model. (C) Example of results from automatic identification of suckling behaviour using the AdaBoostM1 model for Calf1. (D) Performance of the AdaBoostM1 model when the training set size is reduced. The symbols show the mean values and the SD (30 runs).

All pre-selected classification models had a mean global accuracy ≥ 96.93% on unseen data (Table S4). They all identified suckling events with a low False Positive Rate (mean FPR < 1%) except for SVM with polynomial kernels (mean FPR > 1% but not exceeding 2%, Table S4). The likelihood of misclassifying an event as suckling when it should have been classified as non-suckling is thus extremely low for most of the pre-selected models. As shown in Fig. 3A, the SVM with linear kernel model and all decision trees are more located at the bottom left quadrant of the figure, indicating that they had a lower Precision and Sensitivity with regard to suckling, in contrast to AdaBoostM1, GentleBoost, Minkovski metrics-based KNN, Cityblock metrics-based KNN, and Cosine metrics-based KNN models that lay on the top right of the figure. The SVM with polynomial kernel model lays on the top left of the figure, indicating that it had a high Sensitivity but had a low Precision. On the other hand, the Bag or Random Forest model, the SVM with Gaussian kernel model, and the Euclidean metrics-based KNN model lay on the bottom right, indicating that they had high Precision but had low Sensitivity. The AdaBoostM1, an Ensemble classifier, was the model presenting the highest F-score. It had the highest Sensitivity while having at the same time a very good Precision (red circle symbol in Fig. 3A). It was therefore the best model for correctly identifying suckling periods. Examples of classification resulting from the AdaBoostM1 are presented in Figs. 3B and 3C. In these examples, the AdaboostM1 model only misclassified non-suckling blocks as suckling in very rare cases (False Positives, 8 in Fig. 3B and null in Fig. 3C—no red cross). However, it misclassified suckling blocks as non-suckling more often (False Negatives, 34 in Fig. 3B and 6 in Fig. 3C—blue crosses). Nevertheless, it detected most of the real suckling blocks (True Positives, 90 in Fig. 3B and 15 in Fig. 3C—red circles).

As shown in Fig. 3D, while there was typical decrease in the performance of the AdaBoostM1 when we reduced the training set size, the model still detected substantial amount of suckling blocks and had a relatively good precision even when using only 10% of the data as training set (mean at 10:90 split for suckling: sensitivity = 0.49 ± 0.08, precision = 0.79 ± 0.06, F-score = 0.60 ± 0.06, N = 30). However, the results seemed more variable (high SD) when training set was really small (Fig. 3D).

Generalization of the supervised machine learning across individuals

We evaluated the generalization ability of the AdaBoostM1 model, the best identified classifier in the context of suckling identification, using a leave-one-out design. The goal was to assess whether we can use data from other individuals to detect the suckling behavior of a new individual. Given that we had three individuals, there were only three possible splitting combinations: (1) blocks from Calf2 and Calf3 as training set (5,766 blocks, 4.5% suckling and 95.5% non-suckling) and blocks from Calf1 as testing set (2,061 blocks, 2.6% suckling and 97.4% non-suckling), (2) blocks from Calf1 and Calf3 as training set (7,256 blocks, 3.6% suckling and 96.4% non-suckling) and blocks from Calf2 as testing set (571 blocks, 8.6% suckling and 91.4% non-suckling), and (3) blocks from Calf1 and Calf2 as training set (2,632 blocks, 3.9% suckling and 96.1% non-suckling) and blocks from Calf3 as testing set (5,195 blocks, 4% suckling and 96% non-suckling).

Results of the leave-one-out design are presented in Fig. 4. All combinations were more located at the bottom part of the Sensitivity versus Precision plot (open symbols). This indicated that the quantity of the detected suckling blocks was low (low Sensitivity) regardless of the combination. The quality was however still good (relatively high Precision) for the first and second combinations (open square and open diamond symbols). The third combination (open triangle symbol) was on the other hand more located at the left of the plot, indicating that the quality of the classification was bad (low Precision). Globally, it can be said that while some good quality classification can be obtained, it still depended on the combination used.

Figure 4 Sensitivity versus precision plot obtained from the AdaBoostM1 model for the suckling class when using leave-one-out splits on data from three humpback whale calves.

In a leave-one-out split, data from two individuals were used for training while one individual is kept unseen for testing. The plot shows the classifier’s performance when all features were included (open symbols) versus when the features that contributed most to the inter-individual variation in terms of suckling were excluded (blue symbols). For comparison, mean results from 60:40 holdout split (purple symbol) and the worst scenario (red cross symbol) are showed.

Figure 5 Analysis of the inter-individual difference between the suckling blocks of three humpback whale calves.

(A) Result of a Random Forest classification (confusion matrix) of the suckling blocks by individuals when all features were used. (B) Importance of the features in the classification by individuals. A sharp drop of the Gini index is noticeable after the 9th feature (red arrow). (C) Result of a Random Forest classification (confusion matrix) of the suckling blocks by individuals when the first 9 most important features were excluded.

As inter-individual variation of the suckling blocks’ kinematic features may have influenced the generalization ability of the AdaBoostM1, we assessed the model’s performance when the features that contributed the most to inter-individual difference were excluded. We performed Random Forest classification of the individuals to check the existence of inter-individual differences on suckling blocks, analyzed the features’ contribution to the differentiation, and identified a set of features to be excluded in order to reduce the inter-individual difference. The Random Forest algorithm indicated that there was effectively a strong inter-individual difference. Indeed, the classification error rate, indicated by the Out-of-the-bag error rate, was very low (Out-of-the-bag error rate = 3.22%). As shown in the Fig. 5A, the Random Forest algorithm was able to correctly predict to which individual each suckling block belongs to in most of the cases (correspondence between true class and predicted class). The Fig. 5B shows the Gini index of each feature. A high Gini index indicated that the feature played great role in the classification (in other words, in the individual differentiation). In the plot, we noticed a sharp drop in importance after the 9th feature. We thus chose to exclude the first 9 features (Gini index > 4): the maximum depth, mean depth, maximum pitch, minimum pitch, mean pitch, minimum depth, mean speed, minimum speed, and the maximum speed. The Fig. 5C shows the new resulting classification of the suckling blocks by individuals when these features were not included. The classification error rate increased notably (Out-of-the-bag error rate = 22.19%). The Random Forest algorithm confused the individual attribution of the suckling blocks more often (more non-corresponding true class and predicted class), indicating that we successfully reduced partially the inter-individual differences. We observed a substantial increase in performance when we re-ran the leave-one-out design using the restricted list of features (mean FSR, mean absolute roll, minimum absolute roll, minimum depth rate, maximum absolute roll, depth rate variance, maximum ODBA, mean ODBA, mean depth rate, ODBA variance, pitch variance, maximum depth rate, speed variance, minimum ODBA, maximum roll rate, absolute roll variance, depth variance, mean roll rate, depth skewness, ODBA skewness, minimum roll rate, ODBA kurtosis, absolute roll kurtosis, roll rate kurtosis, speed kurtosis, absolute roll skewness, depth rate skewness, speed skewness, pitch skewness, roll rate skewness, pitch kurtosis, depth rate kurtosis, roll rate variance, depth kurtosis) (Fig. 4, blue symbols). Indeed, the Global accuracy was higher than when including all features, and the FPR for suckling blocks decreased. Most importantly, the quantity of detected suckling blocks increased (higher Sensitivity). Also the quality of the classification improved globally (increased Precision), even for the combination that had a very poor Precision at first.

Discussion

Our study characterized the suckling behavior in humpback whale calves less than 3-month old using CATS cam tags on one calving ground in the South Western Indian Ocean. Several important aspects of this vital behavior were investigated: (i) duration and frequency, (ii) occurrence in the water column, (iii) modalities, (iv) behavioral signatures, and (v) the possibility to use only acceleration and depth-derived data (i.e., depth, depth rate, speed, Fluke Stroke Rate (FSR), Overall Dynamic Body Acceleration (ODBA), pitch, roll, roll rate, which were obtained using the data from the depth sensor and/or the 3-axis accelerometer only) to perform an automatic detection of suckling events.

Advantage of using animal-borne camera-equipped multi-sensor tags

Animal-borne multi-sensor tags equipped with video camera are very efficient for studying suckling behavior since they can deliver a view from calf’s perspective and thus confirm evidence of suckling. Video from tags allowed differentiation of events in which the calf started to intentionally touch its mother’s mammary slit to suckle from events when the calf was only positioning itself under the mother. Such differentiation between suckling and non-suckling periods would not be possible if the visual observations were performed from a different perspective or at distance (e.g., surface ship-based, or aerial drone-based), therefore leading to biased estimations of suckling behavior.

In our study, the view offered by the video camera allowed clear definition of suckling events initiated by physical contact between the calf’s snout and the mammary teat. Such contact did not happen every time the calf was under the mother. Suckling behavior was further confirmed by the presence of a milk cloud as in Tackaberry et al. (2020). This method greatly improved describing suckling behavior with confidence.

Previously, positioning and posture of the calves have been used frequently as a proxy for determining the occurrence of nursing or suckling in humpback whales (Glockner & Venus, 1983; Glockner-Ferrari & Ferrari, 1985; Clapham & Mayo, 1987; Morete et al., 2003; Videsen et al., 2017; Zoidis & Lomac-MacNair, 2017). As previously shown by Tackaberry et al. (2020) with older calves (studied in their feeding grounds), we found that young calves were also positioned frequently under the mother and in close proximity but they were actually not suckling. Furthermore, suckling dives (i.e., dives during which suckling was visually confirmed by evidence of a milk cloud) appeared to be much rarer in comparison to non-suckling dives during which the calf was observed under the mother. Consequently, great care must be taken when associating calf positioning with suckling or with suckling dives as these are not necessarily associated. The calf, when positioning under the mother while not necessarily suckling may be taking advantage of reduced drag, as shown for dolphin calves (Noren & Edwards, 2011).

Duration and frequency of suckling events

Suckling events were brief and rare (18.8 ± 8.8 s on average and <2% of observation time). They were shorter in duration compared to those observed by divers in another breeding area (30.6 ± 17 s on average, Zoidis & Lomac-MacNair, 2017). This difference might be related to the method used, as discussed above. The average suckling duration reported in the feeding area is quite similar to what we observed (23 ± 7 s, Tackaberry et al., 2020). The associated suckling frequency was however much lower in the feeding area (0.3% of the time for an observation between 0745 and 1700 local time; (Tackaberry et al., 2020) compared to what we observed in the calving area (1.7% between 1000 and 1800 local time). This expected difference suggests suckling rate varies with age (greater for young calves in the breeding grounds versus older calves in the feeding grounds), as is the case with several species of pinnipeds (Oftedal, Boness & Tedman, 1987).

Occurrences of suckling in the water column

As in Tackaberry et al. (2020), most of the suckling events occurred at depth and during dives. Humpback whales may favor nursing at depth to help the calf in maintaining the suckling posture as the latter would be more buoyant at surface, or for facilitating the thermoregulation of the adult female (Videsen et al., 2017). We noticed that the rare surface suckling events only occurred at the end of the day, when the sun was less intense, suggesting a possible link between surface nursing events and thermoregulation behavior. Further investigations on suckling during the night are needed for comparison, as most videos were recorded during daylight only (this study; Tackaberry et al., 2020). Nursing behavior might show different patterns at night for the reasons cited above. If indeed suckling at night is more common and is occurring more at the surface, this has implications for the risk of boat-strikes at a time when visual cues are limited (for those at the helm).

The maximum suckling depth we recorded (32.6 m) was much shallower than the maximum reported by Tackaberry et al. (2020). However, this does not necessarily relate to any physical limits of young calves. Rather, this maximum suckling depth might be an effect of the environment. Indeed, the average depth of the Sainte Marie channel is 35 m (Trudelle et al., 2016) and in our video recordings it was common to see the seabed when the whales dove to around 30 m depth.

All suckling events during dives occurred during the descent and bottom phases only, unlike in Tackaberry et al. (2020) where they also observed suckling events during the ascent phase. One possible explanation is that very young calves found in breeding grounds have less breath capacities (Saloma, 2018) than older calves found in feeding areas. Thus, during the ascent phase of their dive, their priority would be to reach the surface to breathe rather than to suckle.

Suckling modalities

With respect to posture, as estimated from the video recordings, the calf positioned itself at 30–45° pitch angle relative to the midline of the mother’s body when suckling. This is in agreement with past observations (Glockner-Ferrari & Ferrari, 1985; Zoidis & Lomac-MacNair, 2017). However, the pitch recorded by the tags (pitch angle relative to the horizontal) during suckling was always relatively low on average (<15°). This suggests that the mother is generally oriented facing slightly downward when nursing its calf even during phases other than the descent (during which the mother does not otherwise have to lean downward).

One of the most innovative findings of our study was the calf’s tendency to roll continuously to the side when suckling. Our data confirm the correspondence between the rolling side and the suckled teat in nine events. The calves rolled to the left to suckle on the left teat and rolled to the right to suckle on the right teat. The rolling behavior might be thus related to the anatomy of the mammary gland, in particular the orientation of the nipple as the nipple is extruded from the mammary slit. If each breast is positioned so that the teat is directed at an angle relative to the midsagittal plane, then the calf would be forced to match that angle with its mouth in order to stimulate the nipple to extrude and then grasp the nipple with its tongue. Video recordings showed the calf’s tongue is directed laterally, visible in the oral gape of the mouth on one side of the head. The calf must roll to one side in order to align the lateral aspect of its mouth with the mammary slit (rolling left to align its right oral gape against the mother’s left breast, and vice versa).

There was also the clear pattern of alternation from one side to the other between successive suckling events. Why then must the calf alternate sides? Perhaps the milk supply of each breast is limited, forcing the calf to suckle both sides in order to obtain a sufficient volume for satiation. This would ensure that the mother’s breasts continue to lactate evenly on both sides. Another reason may be related to the milk production/storage/ejection system. It is widely accepted that milk is voluntarily ejected by the mother into the calf’s mouth in cetaceans (Slijper, 1966). It is possible that the amount of milk the mother can continuously eject is limited and the gland may need a refractory period during which it must reset before the next ejection can occur. If so, then alternating which side to suckle maximizes feeding for the calf while increasing milk delivery efficiency for the mother. In this scenario, the currently suckled breast performs milk ejection while the previously suckled breast refills in preparation for the next ejection. This would thus force the calf to alternate between the two mammary glands to get enough milk during the successive nursing bouts. In-depth anatomical investigation is needed to answer this question, as there is scant literature on whale lactation or breast anatomy. The pattern of alternation between the two mammary gland, if indeed necessary and related to the milk production/storage/ejection system in whales, may have an implication on the post-partum viability of twins in cetaceans.

Regarding the anatomy of suckling, it is important to note that in a few cases, rhythmic movements of the calf’s tongue were observed during suckling events. This suggests that the calf actively participates in directing milk into its mouth. It is unclear whether these movements are stimulatory to the mother’s ‘let down’ reflex (nipple erection and milk ejection), serve to ‘strip’ the nipple (as occurs in many land mammals to squeeze milk out), or create a piston-like suction to draw out milk (perhaps in combination with milk ejection from the mother). The lateral aspect of the calf’s rostral tongue is comprised of elongated marginal papillae that may serve to grasp the nipple. This latching-on-nipple function has been proposed for newborn tongues in other whale species (Kastelein & Dubbeldam, 1990; Shindo et al., 2008; Ferrando et al., 2010; Kienle et al., 2015). These marginal papillae may also ‘zipper’ together to form a tube for channeling milk into the mouth, compensating for the calf’s lack of lips and cheeks to seal and contain the nipple and the extruded milk. Alternatively, the rhythmic movements of the tongue may be only an artefact produced by the calf’s swimming movements as the calf undulates its body to maintain its suckling position while the tongue maintains constant contact with the nipple. Further study of the calf tongue may provide additional insights to its function.

Behavioral signatures of suckling events

The comparison of the data derived from accelerometer and depth sensor for suckling and non-suckling events revealed mainly one distinctive characteristic of the suckling behavior: a high absolute roll. Other characteristics, although not obvious, such as high FSR and low speed are also suggested by the results of our comparisons. For suckling occurring during the descent phase, there were additional notable specificities: a low depth rate and a pitch close to zero. This is in sharp contrast with descent non-suckling where calves showed a high descent rate on average and were directed downwards.

To our knowledge, no study to date has highlighted the characteristics of suckling behavior of whales in terms of roll recorded by an accelerometer. The sustained roll deviance from zero during suckling events is directly related to the aforementioned rolling pattern observed in the videos: rolling to the side may facilitates access to a particular mammary teat.

With respect to the kinematics, lower speed while still deploying some physical effort (high FSR) when suckling is consistent with the results of Tackaberry et al. (2020). Even though the pair makes little or no forward movement during suckling, the calf still has to maintain actively the suckling posture to stay in physical contact with the mammary slit. Unlike Tackaberry et al. (2020), we however did not found any specificity of suckling with respect to ODBA. Concerning the descent suckling, the relatively low depth rate may help the calf in maintaining the suckling posture.

Automatic detection of suckling using accelerometer and depth- derived data

In the field of ethology, the advances in machine learning have offered the opportunity to classify behaviors within a complex database (Valletta et al., 2017). Given the stereotyped traits of suckling behavior in humpback whale calves, we tested whether it was possible to apply supervised machine learning algorithms on labelled accelerometer and depth sensor data to automatically discern suckling from non-suckling periods. Although accelerometer data have been already largely coupled with supervised machine learning to detect behaviors in various species (Nathan et al., 2012; Carroll et al., 2014; Ladds et al., 2016; Jeantet et al., 2020), our study is the first attempt to use it for humpback whales and in the framework of a suckling behavior study.

The biggest challenge in the automatic classification of suckling behavior is probably the class imbalance. Indeed, suckling events are naturally rare (this study; Tackaberry et al., 2020). It is argued that unbalanced data tend to bias predictions in favor of the majority class. This bias is problematic in situations in which missing the minority class case is worse than misclassifying a majority class (Leevy et al., 2018). In the case of suckling versus non-suckling classification, this bias is not a significant concern since there is more interest in minimizing the False Positive Rate (FPR) for the targeted behavior (i.e., suckling). Such conservative measure is generally adopted in behavioral detection (Nathan et al., 2012; Carroll et al., 2014; Tennessen et al., 2019). On the other hand, the best model we found for identifying suckling periods, the AdaBoostM1, is a model suitable for unbalanced binary classifications (Galar et al., 2011; Leevy et al., 2018). Thus, class imbalance is less of a concern. In order to avoid evaluation bias, the evaluation process was guided primarily by the minimization of the FPR rather than the Global accuracy, since the latter does not distinguish between the numbers of correctly classified examples of different classes and can be misleading in the framework of an unbalanced dataset (Galar et al., 2011). It is important to recall that we also performed a data segmentation (we excluded blocks at <1.5 m depth or at >2 m s−1 speed) that helped in reducing data imbalance and in improving the results. While the thresholds we chose for the segmentation works well with our data and are compatible with our current knowledge on the suckling behavior of humpback whale, they may be subject to change and adjustment as our knowledge on humpback whale behavior evolves.

With an initial 60:40 holdout training-testing split, the average Sensitivity and Precision of the AdaBoostM1 model with respect to suckling class were of 0.71 and 0.89 respectively and the mean FPR was <1%. In other words, the model was able to detect the vast majority of suckling blocks and about 9 out of 10 blocks reported as suckling corresponded to visually confirmed suckling periods.

The Leave-one-out design (blocks from two individuals as training set and the one remaining individual as a testing set) allowed testing whether data from two individuals can be used to detect suckling behavior in a novel individual. In this design, it was expected that the performance would drop notably (lower Sensitivity and Precision). Indeed, the inter-individual difference has been shown to generally penalize the performance of automatic classification models (Vázquez Diosdado et al., 2015; Ladds et al., 2016). When we reduced the inter-individual difference in terms of suckling by excluding features that introduce substantial inter-individual differences, the performance returned to a reasonable level regardless of the combination (i.e., the model generalized better and became more robust). Indeed, out of every 10 blocks reported as suckling, at least about 6 blocks were confirmed to be correct (lowest Precision = 0.59). The associated Sensitivity was relatively low, indicating that several suckling blocks were not detected. However, detecting only few blocks but with precision is acceptable in the case where the goal is to be sure that a positive result means the identified suckling blocks really correspond to suckling periods. Furthermore, with only few blocks detected with high confidence, complete suckling events (start and end) may be determined by examining the corresponding sustained roll deviations, thus compensating the lack of sensitivity.

We were not able to investigate the effect of individual tag differences due to limited sample. We can however expect that this effect, if any, is relatively small. In the first and second combinations of our leave-one-out experiments, the two calves used for training were tagged with different individual tags (Cats1 and Cats2) and the remaining calf used for testing was tagged with one of the individual tags involved in training (Cats1 or Calf2). However, in the third combination, the two calves used for training were tagged with the same individual tag (Cats1) and the remaining calf used for testing was tagged with a different individual tag (Cats2). The fact that there was no evident divergence of the results of our third combination from the results of the first and second combinations suggests that our model was not significantly affected by the differences in individual tags. Further studies are encouraged regarding the potential effect of individual tag differences since our conclusion is still limited by sample size.

As more data is added to the learning set, we can expect better performance from the models. Indeed, we showed that the size of the training set can slightly influence the accuracy of the results. Similarly, we can also expect that with more individuals, the constraints and restrictions related to inter-individual differences would also be reduced (Ladds et al., 2016).

These results on automatic identification of suckling events are very promising for rapid evaluation of larger data sets and will open new opportunities for investigating limited data sets. First, the model may be adjusted to accommodate for incomplete observations (altered/missing video file, night-time recording, etc.) using only two sensors (3-axis accelerometer and depth sensor). Secondly, the model may be applied to data from comparable multi-sensor tags that lack visual support (video recordings lacking or too dark, tags without camera such as Acousonde and Dtags, etc.).

While these methods will broaden the opportunities in the domain of humpback whale mother-calf behavioral studies (and eventually may be applied to studying other large whales), care must be taken in these early stages. Until the acquisition of large training data occurs (several individuals), researchers should be aware of the limitations of each trained model. For example, possible differences in gaits and behavioral pattern with respect to age (Noren, Biedenbach & Edwards, 2006; Saloma, 2018) may hinder the generalization ability of the models. For our trained model, we caution that it should be conservatively applied to less than 3-month old non-neonate calves occurring in a shallow water. Nevertheless, a concrete application example of the automatic identification of suckling behavior would be the investigation of the behavioral time budget of mother-calf pairs over 24 h (day and night). The main difficulty in using multi-sensor tags for studying the behaviors of mother-calf pairs concerns the acquisition of sufficiently representative data. Indeed, tagging whales is a challenge per se and some age classes are more difficult to observe (e.g., neonate calves are rarely observed). Moreover, the tag attachment time on a whale is often limited (few hours on average), thus it is rare to record behaviors of individual whales over long periods (>24 h). Future studies should address the ontogeny aspect of calves’ behavior in order to subsequently assess its implication in the field of automatic behavior detections. The development of more durable yet non-invasive tag attachment systems is also highly encouraged as this is the key to acquiring more complete data and thus obtaining more representative learning set for accurate automatic behavior detection.

Conclusions

Our results provide new knowledge on the suckling behavior of humpback whale calves in their calving ground. Our descriptions, based on videos from the calf’s perspective, accelerometer data, and depth data complement previous studies based on surface and sub-surface observations (Glockner & Venus, 1983; Glockner-Ferrari & Ferrari, 1985; Clapham & Mayo, 1987; Morete et al., 2003; Videsen et al., 2017; Zoidis & Lomac-MacNair, 2017). We found that suckling is more frequent compared to what has been reported in the feeding area (Tackaberry et al., 2020), suggesting a variation of the suckling rate with the calf’s age and underlining the importance of the suckling behavior in the calving area. We also confirmed that most suckling events occur at depth, highlighting the importance of using multi-sensor tags equipped with a video camera in order obtain reliable observations. Finally, we found that suckling behavior is stereotyped, especially characterized by a continuously sustained roll deviating significantly from zero and likely a high FSR and low speed, and can potentially be detected automatically by supervised machine learning. While our results are only based on three tag deployments on humpback whale calves in the breeding ground, they are a proof of method on which future investigations of suckling and nursing behavior in humpback whales can build upon. Our study should be replicated with other baleen whale species for which suckling behavior is still undiscovered.

Supplemental Information

Supplemental Information 1 Supervised machine learning workflow for automatic identification of suckling behavior in free ranging humpback whale calves using CATS cam tags data

The data derived from the 3-axis accelerometer and the depth sensor of the tags, sampled at 10 Hz, was split into 2 s non-overlapping blocks (windows) and each block was labeled either as ‘suckling’ or ‘non-suckling’ depending on which behavioral period it fell under (suckling period or any non-suckling period). The data underwent segmentation in order to partially remove noise and also reduce the class imbalance. The thresholds we used (< 1.5 m depth and > 2 m s–1 speed) were based on the known characteristics of the nursing/suckling behavior of humpback whales and their validity in our datasets was checked to ensure that no suckling events were removed partially or entirely following the segmentation. The pie charts represent the class distribution (suckling versus non-suckling). N = 18, 331 and N = 7,827 before and after segmentation respectively.

Click here for additional data file.

Supplemental Information 2 Depth, pitch, roll, speed, FSR and ODBA during and around suckling events for each calf

Blue, red, and green areas in the depth profile correspond to the descent, bottom, and ascent phases of dive respectively. Uncolored areas in the depth profile correspond to surface phases. Suckling events are identified in red in the depth profile and by yellow box in the raw pitch(blue)/roll(yellow) profiles. Yellow box placed on top indicate that the calf was observed rolling to the right side on the corresponding video. Yellow box placed on the bottom indicate that the calf was rather observed rolling to the left side. Events during which the teat suckled by the calf was clearly identified on the corresponding video are marked with (R) or (L): (R) for right teat and (L) for left teat.

Click here for additional data file.

Supplemental Information 3 Temporal distribution of suckling events through the duration of the deployment for three tagged humpback whale calves

Blue, red, and grey bars correspond to Calf1, Calf2, and Calf3 respectively. Tag detached at 164 min for Calf1 and at 98 min for Calf2. For Calf3, the tag detached at 521 min. However, only the first 479 min (92%) of the data has been analyzed due to lack of visibility on the video recording as the evening approached. There was no evidence of any trends with suckling frequency increasing later in the deployments. The periods towards the end of the deployments were not necessarily associated with more frequent suckling events.

Click here for additional data file.

Supplemental Information 4 Comparison of the number of non-suckling dives during which the calf was observed staying in close proximity beneath the mother for at least five consecutive seconds and the number of suckling dives

Click here for additional data file.

Supplemental Information 5 Summary table of the mixed effect models for the characteristics of suckling events

The models included the suckling status and activity phase (descent, bottom, ascent or surface) as fixed effect and individuals as random effect (reference level = bottom and non-suckling, i.e., bottom non-suckling). The data used in the analysis consisted of individual suckling events (18.8 s average duration) and random assortments of non-suckling segments (20 s duration). Significant P (<0.05 in this study) are marked in bold. CI: Confidence interval. SE: Standard Error of estimate.

Click here for additional data file.

Supplemental Information 6 Comparison of suckling events (18.8 s average duration) and non-suckling segments (20 s duration) with respect to activity phase using Tukey’s post-hoc multiple comparison test

Comparisons were computed using the R package emmeans. Significant P (<0.05 in this study) are marked in bold. SE: Standard Error of estimate.

Click here for additional data file.

Supplemental Information 7 Results of the machine learning optimization procedure for identifying automatically suckling in non-labelled data

A Bayesian optimization procedure was run by type of model (Ensemble, KNN, Decision tree, SVM) 30 times in order to select the best model (by seeking to minimize classification error). Values are presented following the format mean ± SD. n represents how often each model was selected by the optimization procedure for each type of classifier over the 30 runs. FPR: False positive rate.

Click here for additional data file.

Supplemental Information 8 Raw data containing characteristics of suckling events (18.8 s average duration) and non-suckling periods (20 s duration) obtained from CATS cam tags deployed on three humpback whale calves in the Sainte Marie channel, Madagascar

Each line corresponds to either a suckling event or a 20 s non-suckling periods selected randomly. The corresponding metadata is provided.

Click here for additional data file.

Supplemental Information 9 Raw data from CATS cam tags deployed on three humpback whale calves in the Sainte Marie channel, Madagascar used for testing automatic identification of suckling behavior using supervised machine learning

Lines correspond to 2 s non-overlapping sliding blocks (windows). The corresponding metadata is provided.

Click here for additional data file.

Supplemental Information 10 R scripts for analyzing the behavioral signatures of suckling events and inter-individual difference between suckling blocks

Click here for additional data file.

Supplemental Information 11 MATLAB scripts for tuning and training supervised machine learning algorithms for automatic identification of suckling behavior

Click here for additional data file.

We would like to warmly thank Aina F. Ramanampamonjy, Léo Duperret, Mandrindra O. Rakotovao, and Nyal Mueenuddin who contributed to the data collection. We warmly thank Conor Ryan, Jennifer Tackaberry, and one anonymous referee for their valuable and constructive comments on the manuscript.

Additional Information and Declarations

Competing Interests

Author Contributions

Animal Ethics

Field Study Permissions

Data Availability

The authors declare there are no competing interests.

Maevatiana N. Ratsimbazafindranahaka conceived and designed the experiments, performed the experiments, analyzed the data, prepared figures and/or tables, authored or reviewed drafts of the paper, and approved the final draft.

Chloé Huetz, Anjara Saloma, Olivier Adam and Isabelle Charrier conceived and designed the experiments, performed the experiments, authored or reviewed drafts of the paper, and approved the final draft.

Aristide Andrianarimisa and Joy S. Reidenberg conceived and designed the experiments, authored or reviewed drafts of the paper, and approved the final draft.

The following information was supplied relating to ethical approvals (i.e., approving body and any reference numbers):

All methods and approaches were fully approved by the Ministry of Fisheries Resources, Madagascar (National research and collect permits #28/18-MRHP/SG/DGRHP and #36/19-MAEP/SG/DGPA).

The study complies with the European Union Directive on the Protection of Animals Used for Scientific Purposes (EU Directive 2010/63/EU).

The following information was supplied relating to field study approvals (i.e., approving body and any reference numbers):

Field studies were fully approved by the Ministry of Fisheries Resources, Madagascar, under the national research and collect permits #28/18-MRHP/SG/DGRHP and #36/19-MAEP/SG/DGPA.

The following information was supplied regarding data availability:

The raw data are available in the Supplemental Files.

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
