# Peer review of "Characterizing the suckling behavior by video and 3D-accelerometry in humpback whale calves on a breeding ground"

_PeerJ, doi:10.7717/peerj.12945_

## Round 0.1 · original submission · Major Revisions

Overall, the manuscript is well written but does require some revisions. I suggest leaving in the section on suckling/nursing in the Introduction. I understand why the reviewer suggested that that text might not be needed, but it is remarkable how many papers use terms like nursing and suckling incorrectly. Please provide a detailed response to all the reviewer comments.

·

Basic reporting

The manuscript has been carefully and fluently written. The aims and results are clearly articulated. It follows a logical structure and is well referenced. There is room for some minor shortening, as I have indicated with tracked changes and there are couple of very minor typos. It was a pleasure to read.

Experimental design

This work fits the scope and aims of the journal and it fills some important knowledge gaps. There is room to briefly expand the relevance of the findings in the discussion (e.g. regarding threats from ship-strikes), as I have indicated using tracked changes. The ethical standards are difficult to assess from the minimal details presented. As I have mentioned, in the tracked comments, more information should to be provided about ethical approval (in line with the standards for this journal). Furthermore, information about the response of the animals to the tagging procedure should be reported.

Validity of the findings

The work appears to be of a high technical standard and I admit that I am not sufficiently familiar with machine-learning to critically review the statistical methods. The conclusions are concise and well reasoned from the results.

Reviewer 2 ·

Basic reporting

Line 367: The meaning of “N” here is inconsistent with that in Line 281 and Line 332. Please use different symbols to represent different meanings or you could describe them using text directly.

Line 370: In Table 1, it’s better to explicit the detailed phases of dives, i.e., changing “during dive” into “during dive (descent)” or “during dive (bottom)”.

Line 394: I believe you want to refer to “Table 3” not “Table 2”. Please check it in the subsequent content.

Line 437-441: The contents should be placed in the methods section.

Line 441: In Figure 5, please unify the expression of runs and iterations, i.e., using either runs or iterations.

Line 465: “training set was small” not “test set was small”.

Line 523-525: Could you first explain what’s meaning of the depth-derived data? What does it include? It seems that the currently used features for machine learning contain all the features (Line 226-228), not limited to the accelerometer and depth-derived data.

The text inside figures is not consistent, and it should be consistent with the manuscript text size and style.

Experimental design

Line 263-265: Preliminary data processing has been conducted, i.e., removing some unwanted blocks. But, as we all know, the collected data might contain some noise resulting from some device or environmental factors during the data collection process. I suggest that you conduct some denoising operations such as signal filtering before feature extraction. It would be beneficial for performance improvement.

Line 277-278: The authors only trained three types of supervised machine learning methods and did not explain why to use them. I suggest that you try a support vector machine and random forest algorithm, which might obtain higher performance.

Validity of the findings

Line 444-446: Please add the detailed results of global accuracy and False Positive Rate for all of the pre-selected models. You could add an external table or figure to show them.

Line 506-514: In Figure 6, it looks that it doesn’t work well for the second combination when using the restricted list of features. I suggest the authors add some explanations. In addition, the sensitivity is a little low and it may be improved if you try some pre-processing method or try other classifiers like SVM. You can also elaborate more about the reason why it’s relatively low (Line 686-687).

Line 686-689: The reason why the priority goal is to maximize precision instead of sensitivity is not clearly explained. It needs further explanation based on the practical significance.

Line 710-712: I have concerns about the main difficulty when applying the automatic identification method in practice? The authors should clearly state the limitations of the proposed method in practical applications and should mention them in the discussion section. In addition, it would be better to add some future works.

Additional comments

Overall, the manuscript is well and detailly written. Looking forward to seeing you with the revised version.

·

Excellent Review

This review has been rated excellent by staff (in the top 15% of reviews)
EDITOR COMMENT
Very comprehensive and helpful review.

Basic reporting

Overall, I think this is a well-done study and well-written manuscript. This work nicely builds on past studies about nursing/suckling behavior in humpback whales and improves our knowledge using modern technologies. It also lays the foundation to automate the detection of suckling events in biologger data. The introduction provides a solid background and establishes the need for this study. However, a few citations are needed to support some of the statements made in the introduction. I provided comments in the "Line by line comments" section on the attached document.

The figures and tables provided are valuable and relevant to include in the manuscript. I have provided specific comments for some of them in the "Figures and Tables" section of the attached document. Overall, I would suggest making sure the captions provide enough information to stand alone from the manuscript. A few figures and tables also need additional labeling to specify which data was actually plotted/reported (mean, maximum, minimum, maximum mean, or minimum mean).

The raw data is not referenced in the main text. If you intend to provide the data with your study, additional metadata (maybe I missed it) tied to the supplemental files (definitions of the column headers and units) would be helpful.

There are a few overall minor suggestions to increase clarity and improve flow. You can choose to use them or not, but I do not think it is necessary for you to address each individual suggestion/comment in that section. I listed some overarching comments (1-3) and provided line-by-line suggestions at the end of the attached document under the headline "Minor suggestion."

Experimental design

This study clearly defines the research questions, the knowledge gap, and how this study has helped to close that gap. I have provided comments in the "Line by line comments" section that I think will allow others to replicate this study more accurately by asking you to include additional details in the methods section. There is an additional need for you to clarify the reporting of your results and provide a little more information as to why you chose the experimental design you used. Since this study is proof of concept for automating detection for humpback whale suckling behavior, I foresee many other studies basing their experimental design on your paper. Therefore, future studies will be improved if you leave little room for interpretation in your protocols and explain why you chose the methods you used.

Validity of the findings

Despite the small sample size, I think this is an important study to publish and introduce to the broader scientific community- for others to build upon and provide data to strengthen the models. I have provided comments in the "Line by line comments" section on the attached document that should lead to improved clarity of the discussion and ensure that the discussion and conclusions reflect and acknowledge the methods and limitations of this study.

---

## Round 0.2 · accepted · Accept

Thank you for the detailed and careful revisions to this interesting study. I am now happy to accept, subject to a final few minor revisions (see reviewer comments).

One of our reviewers is travelling with intermittent internet access, so they emailed their comments: "Comments: the authors did an excellent job of addressing the concerns of the reviewers and the manuscript has been improved. It will be a very important contribution to our knowledge on whale suckling behaviour."

Reviewer 2 ·

Basic reporting

The authors have addressed all the issues I pointed out before, and the latest manuscript has been clearly and well written. However, references to newer or more recent work are required since the current paper mostly references are outdated. It is more meaningful to provide newer ones. In addition, some sentences are too long (e.g., Line 107-111, and Line 528-532), which easily confuses readers. It is better to rewrite in two shorter ones.

Experimental design

The authors have added more details about the analysis process of the raw data and compared several popular machine learning methods sufficiently.

Validity of the findings

This study gave completed results and reasonable discussion, and also pointed out its limitations and future works. It’s very useful for further investigation on this kind of researches.

Additional comments

Overall, this manuscript has been well done. I have no opposing opinion for publishing this paper.

·

Basic reporting

I believe the authors have thoroughly addressed any comments or concerns I had regarding the original manuscript. The paper reads well, and it appears that the authors have included additional text, citations, and analysis to improve the study's clarity, content, and replicability.

Experimental design

The authors clearly stated the aims of the study and the knowledge gaps they were addressing. The analysis completed within my scope of experience appears to meet high technical standards- I defer to the other reviewers to comment on the additional analysis regarding automated identification and machine learning.

Validity of the findings

The authors provided underlying data and encouraged future studies to strengthen their findings. They also addressed the impact and novelty of their study. I agree that this is an important study to further our understanding of humpback whales' suckling behavior and further our ability to use biologgers to study marine mammal behavior.

Additional comments

I believe there is a typo on line 428. It currently reads, "Additional accelerometry singles, ..." I am guessing the word is meant to be "signals". Please ignore if I am mistaken.